# Recent Progress on Hyaluronan-Based Products for Wound Healing Applications

**DOI:** 10.3390/pharmaceutics14102235

**Published:** 2022-10-19

**Authors:** Kuncham Sudhakar, Seong min Ji, Madhusudhana Rao Kummara, Sung Soo Han

**Affiliations:** School of Chemical Engineering, Yeungnam University, 280 Daehak-Ro, Gyeongsan 38541, Gyeongbuk, Korea

**Keywords:** hyaluronic acid, metal/metal oxide nanoparticles, antibiotic drugs, antibacterial activity, wound healing

## Abstract

Hyaluronic acid (HA) based nanocomposites are considered excellent for improving wound healing. HA is biocompatible, biodegradable, non-toxic, biologically active, has hemostatic ability, and resists bacterial adhesion. HA-based nanocomposites promote wound healing in four different sequential phases hemostasis, inflammation, proliferation, and maturation. The unique biological characteristics of HA enable it to serve as a drug, an antibacterial agent, and a growth factor, which combine to accelerate the healing process. In this review, we focus on the use of HA-based nanocomposites for wound healing applications and we describe the importance of HA for the wound healing process in each sequential phase, such as hemostasis, inflammation, proliferation, and maturation. Metal nanoparticles (MNPs) or metal oxide nanoparticles (MO-NPs) loaded with HA nanocomposite are used for wound healing applications. Insights into important antibacterial mechanisms are described in HA nanocomposites. Furthermore, we explain antibiotics loaded with HA nanocomposite and its combination with the MNPs/MO-NPs used for wound healing applications. In addition, HA derivatives are discussed and used in combination with the other polymers of the composite for the wound healing process, as is the role of the polymer in wound healing applications. Finally, HA-based nanocomposites used for clinical trials in animal models are presented for wound healing applications.

## 1. Introduction

Skin is the first-line anatomical barrier, the most exposed organ, and the most vulnerable to injury. Skin is a primary immunological barrier and protects against dehydration and infiltration by micro-organisms. When skin is damaged, complex biological processes are immediately initiated comprised of four overlapping, sequential, interconnected phases, namely, (i) hemostasis and coagulation, (ii) inflammation, (iii) proliferation and migration, and (iv) maturation or remodeling [1,2,3,4,5]. The majority of wounds are easily cleaned and recover within a few weeks with appropriate care. However, in many individuals with injuries or more serious wounds, healing is prolonged becoming problematic. These complications include hemorrhage and infections and increase treatment costs and have negative consequences for individuals, their families, communities, and society [6,7,8]. Wounds are a serious global problem, and associated treatment costs were projected to be US $20 billion and €4–6 billion annually in the United States and the European Union, respectively [9,10,11]. Skin lacerations caused by physical, chemical, and thermal factors initiate the wound healing process. However, wounds may become chronic as a result of loss of fluids, excessive inflammation, and other hurdles that prevent or delay wound healing [12,13,14]. To circumvent these limitations of the wound healing process, tissue engineers have developed various biomaterials, such as hydrogels, membranes, fibers, and nanocomposites. Ideal wound dressings must be able to: (i) preserve a wet environment; (ii) facilitate epidermal migration, promote angiogenesis, and organize connective tissue; (iii) permit access to gases and nutrients; (iv) protect against micro bacterial contamination; and (v) be non-toxic, biodegradable and non-allergenic [15,16].

Wound healing is an interactive process that includes phases of hemostasis, inflammation, proliferation, and remodeling. In the wound healing complex, the most serious problem in skin wounds is a bacterial infection, which can be caused by the aggregation and expansion of bacterial cells at the wound site [17,18]. Infection under the dressing prolongs the inflammatory response, impedes re-epithelialization and collagen synthesis, and inhibits healing [19]. The most common bacteria accompanying wound infections contained Gram-positive and Gram-negative bacteria. Moreover, bacterial cells can also aggregate and anchor in an extracellular matrix to form biofilms. As a result, they can prevent drugs from penetrating the affected region, and bacterially formed biofilms are considered an important mechanism of antibiotic resistance against bacterial infections [20]. Bacterial infections at wound sites have been associated with delayed wound healing and, in certain circumstances, with significant adverse consequences that can be life-threatening. It follows that antimicrobial activity is an essential requirement for an appropriate wound dressing. Some studies have been reported on the importance of antibacterial activity for skin re-epithelization during the wound healing process [21,22,23,24].

## 2. HA Plays a Significant Role in the Wound Healing Process

HA is a linear glycosaminoglycan (GAG) present in all living creatures including bacteria and has a consistent chemical structure. Alternating repeating units of D-glucuronic acid and N-acetyl-D-glucosamine are connected by β-1, 4, and β-1, 3-glycosidic associations in biopolymeric HA [25]. HA is a constituent of dermal extracellular matrix (ECM), and is present in numerous tissues, including synovial fluid and soft and connective tissues [26,27]. HA has a rapid turnover rate in vivo and is catabolized in the extracellular environment via two pathways: (a) by hyaluronidase hydrolysis or by oxygen-free radical fragmentation before being evacuated from the lymphatic system, and (b) by enzymatic degradation initiated by adjacent cells, after which it is transported to lysosomes or endosomes [28]. The biological function of HA is reliant on molecular weight (MW) [29,30]. Specifically, HA receptors, such as CD44 (cluster of differentiation 44), RHAMM (receptor for HA-mediated motility), and TLR2 and 4 (toll-like receptors 2 and 4), interact with HA of different molecular weights [31], and these interactions regulate wound healing through several intracellular signaling pathways, though wound healing usually proceeds via the four sequential stages shown in Figure 1.

Phase I—Hemostasis and coagulation: Platelets within wounds generate significant amounts of high MW HA (HMW-HA). Edema occurs when HMW-HA is linked to fibrinogen, clotting factor I, which circulates in blood and causes effective clot formation. Edema forms when HA is saturated with fluid and HA expands around wound sites. Furthermore, immune cells access wounds through edema and provide a temporary scaffolding material [32,33];Phase II—Inflammation: HMW-HA is converted into low molecular weight HA (LMW-HA) at sites of inflammation, and whereas HMW-HA has immunosuppressive and antiangiogenic properties, LMW-HA is immunostimulatory and pro-angiogenic. Cytokines and chemokines induced by LMW-HA bind to TLR2 and TLR4 at wound sites and promote the activation, infiltration, and maturation of immune cells. However, although inflammation is essential for wound healing, protracted inflammation can cause acute wounds to become chronic. During the Inflammation, LMW-HA is converted into oligomer-HA (O-HA), which suppresses inflammatory response and increases proliferative activity [34];Phase III—Proliferation and migration: In this phase, O-HA reduces inflammation, increases re-epithelization, promotes angiogenesis, and accelerates granulation tissue development. O-HA binds to CD44 and RHAMM consequently activates keratinocytes, endothelial cells, and fibroblasts. Furthermore, to produce and deposit collagen type III at wound sites, O-HA promotes endothelial and fibroblasts, which results in the formation of new collagen matrix [35];Phase IV—Maturation and remodeling: This phase is characterized by interactions between O-HA and the receptors CDD4 and RHAMM, which result in the production of type I collagen. ECM remodeling requires the upregulations of matrix metalloproteinases (MMPs) and transforming growth factor (TGF-β), which stimulate fibroblast to myofibroblast differentiation. Therefore, HA-enriched composite facilitates the great potential for wound regeneration [36]. Table 1 shows the summary of the wound healing process.

## 3. Wound Healing Applications of Metal Nanoparticles (MNPs)

It is essential that clinicians anticipate the needs of injured tissues. Material choice is highly influenced by the phase of healing, duration of the therapeutic effect, dosage, wound depth, and mechanism of action [37]. Researchers have created materials that maintain a moist environment and be antimicrobial. Individual wound healing routes have been constructed using a range of natural and synthetic materials alone or in combination [38]. Furthermore, nanomaterials have great potential in terms of encouraging self-healing processes that imitate regeneration. However, due to the heterogeneous nature of wound tissues, an understanding of the mechanisms and cellular cascades involved is required to customize nanomaterials for wound healing applications [39,40]. Nanoparticles are intrinsically more active than microparticles because of their higher surface-to-volume ratios. Metal nanoparticles, such as Ag, Au, and Zn, have antimicrobial activities and remarkably stimulate wound healing, making them suitable candidates for wound dressings [41,42,43].

Silver nanoparticles (AgNPs) can regulate the release of anti-inflammatory cytokines and speed wound healing without causing scarring, and encourage keratinocyte proliferation to promote epidermal re-epithelization [44]. Gold nanoparticles (AuNPs) improve healing and limit microbial colonization and increase keratinocyte proliferation and differentiation at low concentrations [45]. Zinc oxide nanoparticles (ZnO NPs) are established antibacterial agents that induce bacterial cell membrane perforation. When nanoparticles are integrated into hydrogels used as wound dressings, they promote keratinocyte migration and thus improve re-epithelialization [46]. On the other hand, polymeric nanoparticles, such as chitosan, alginate, cellulose, and hyaluronic acid, have strong antibacterial and re-epithelialization characteristics when used as wound dressing materials or delivery vectors. Hyaluronic acid modulates cell adhesion and attachment [47], and hyaluronan oligosaccharides induce endothelial cell proliferation, motility, and angiogenesis by increasing the production of vascular endothelial growth factors [48].

Nanoparticle structure-activity correlations and the mechanisms responsible for their actions have remained elusive. However, nanomaterials offer a great deal of promise as they promote self-healing mechanisms that mimic regeneration. Understanding of the processes and cellular cascades at play to design nanomaterials that promote wound healing [37]. This section describes HA interactions with antibacterial agents and their antibacterial mechanisms. The interaction between HA and MNPs with inter and intra-hydrogen bonds between hyaluronan chains is destroyed or generated spontaneously when the pH of the reaction mixture is increased or decreased. [49]. MNPs are chemically or physically attached to hydroxyl, carboxyl, acetamide, and the amino groups of hyaluronan. The hyaluronan groups electrostatically protected MNPs which MNPs function as crosslinking agents between hyaluronan chains and maintain the helical structure of hyaluronic acid [50]. The rapid development of resistance mechanisms in several bacterial strains makes it difficult to combat antibiotic-resistant pathogens [51]. In individuals with some degree of immunosuppression, these strains are particularly capable of causing bacteremia and persistently infected wounds [52]. By damaging the cell wall, bacterial membranes, electron transport chain, nucleic acids, proteins or enzymes, these inorganic nanoparticles can prevent bacterial growth [53]. They can also do this by binding directly to biological macromolecules or indirectly producing reactive oxygen species (ROS). However, under physiological circumstances, using the higher amounts of MNPs required for antibacterial activity can result in poor biocompatibility and cytotoxicity, modification of the MNP surface with additional biocompatible antibacterial compounds is a promising strategy [54]. In order to understand the wound healing process, it is necessary to understand the antibacterial mechanism.

### Mechanisms of HA Nanocomposite Induced Releases of Antibiotics and MNPs 

The mechanism underlying the antibacterial activity of HA nanocomposites is shown in Figure 2. The main steps are as follows: (a) Gram-positive bacteria and Gram-negative bacteria released hyaluronidase (HAase), which causes specific cleaved of the HA-Ag-drug nanocarrier [53,54]. Furthermore, metabolic activities of the antibacterial agent cause an acidic environment (H^+^) in the vicinity of bacteria [55]. Both HAase and H^+^ may cause MNPs and drugs released from HA composite. (b) MNPs have been shown to adhere and accumulate on cell membranes, where they may oxidize plasma membrane surface proteins and cause structural alterations in cell membranes [56]. (c) However, the M^+^ produced by MNPs dissolves cell membranes and causes considerable increases in membrane permeability and morphological changes [53,57]. (d) MNPs and M^+^ have also been shown to produce ROS, which can cause cell membrane damage [58]. (e) Membrane leakage can reduce the transmembrane proton electrochemical gradient and disable energy-dependent processes, such as adenosine triphosphate (ATP) production, ion transport, and metabolite sequestration [59]. (f) MNPs can easily enter bacterial cells by enhancing cell membrane permeability and then biochemically attach to ribosome at h44 in the 30S and H69 in the 50S subunit and prevent protein formation [60]. (g) Damaged DNA caused by MNPs can interact with intracellular enzymes and disrupt cellular metabolism-induced intracellular ROS production [54,61,62]. (h) Furthermore, interactions between MNPs and antibiotics produce MNPs-antibiotic complexes, which stimulate M^+^ ion release at cell walls and inhibit bacterial growth [63]. A synergistic antibacterial mechanism in which MNPs-antibiotic complexes increased ROS levels, membrane damage, and K^+^ leakage as a result of protein release and biofilm inhibition [64].

## 4. HA-MNPs and Wound Healing Applications

### 4.1. Silver Nanoparticles

Silver (Ag) is the most well-studied antibacterial agent due to its broad antibacterial spectrum, although extensive spectra of micro-bacterial pathogens, fungi, and viruses are resistant to AgNPs [65,66]. Since World War I, silver nitrate (AgNO_3_) has been used as an antibacterial material for treating wounds because of its unique antibiotic properties. However, the current use of Ag agents for the treatment of burns is limited to silver derivatives, such as silver sulfadiazine (cream/gel) [67]. AgNPs play a vital role in preventing infection and decreasing bacterial effects at wound sites due to their broad-spectrum antibacterial and surface modification characteristics, and thus, are incorporated into polymeric materials and drugs to promote wound healing [68]. However, antibacterial agents must be added to native hyaluronan to prevent bacterial infections of wounds or chronic ulcers from forming. Table 2 shows summarize MNPs-loaded HA composites used for wound healing applications.

In-situ AgNPs were synthesized using HA as a stabilizing and capping agent without any external reducing agents by A.M. Abdel-Mohsen et al. [69]. The in-situ production of AgNPs (20–25 ± 2 nm) enhanced the crystallinity and thermal stability of fabrics and improved mechanical characteristics as compared with pure HA. Wound covers made of HA/AgNPs nanocomposite fabrics had satisfactory antibacterial activities against Gram-negative bacteria (e.g., *E. coli*), did not show cytotoxicity against HaCaT cells, and were highly biocompatible. HA/AgNPs composites also had better wound healing efficacies than a HA composite and a control sample. Histological evaluations confirmed the improved wound healing capacities of HA/AgNPs composites. 

B. Lu et al. [70] reported spongy composites (SPCs) containing AgNPs prepared from HA combined with a chitosan solution mixed and AgNO_3_ using a freeze-drying method. AgNO_3_ concentration was found to influence AgNPs aggregation and morphology. HA/AgNP (5–20 nm) SPCs had an interconnected porous structure and a rough surface. When AgNO_3_ concentration increased, pore size increased, and surfaces had a folded structure. SPCs had good mechanical properties, swelling, and water retention capacity (>5% after 60 h). Furthermore, their results indicated that SPCs could effectively inhibit bacterial growth and the penetration of *E. coli* and *S. aureus*. L929 cells were unaffected by a low concentration of AgNPs. In vivo studies showed wound contraction ratio, average healing time, and histological characteristics that HA/AgNPs composites encouraged wound healing. Figure 3 shows the formation of AgNPs and bacterial killing for wound healing application.

P. Makvandi et al. [71] fabricated injectable-sensitive hydrogels based on a combination of HA, pluronic (P), corn silk extract (CSE), and AgNPs using a microwave-assisted method. The gelation of pluronic is induced by self-assembly triggered by polymer-polymer interactions in solution to form a semi-solid phase [72]. AgNPs increased the population of micelles and increased their sizes in composite gel, but HA did not affect the size of micelles. HA/P/AgNPs hydrogels exhibited good mechanical properties with T_gel_ values close to body temperature and improved viscoelasticities. Fabricated AgNPs (13 ± 1 nm) hydrogels effectively inhibited *B. subtilis*, *S. aureus*, *P. aeruginosa*, and E. coli growth, and L929 cells treated with AgNPs exhibited no evidence of cytotoxicity and a typical mouse fibroblast-like shape typical of in vitro L929 cell morphology shown in Figure 4A(a,b) for 24 h and 72 h. HA/P/AgNPs hydrogels were applied to HDF cells for in vitro wound healing assessments. CSE acts as a medium for synthesizing AgNPs and improving wound healing ability. Pluronic can promote cell migration, division, and proliferation, increase ECM formation, and help to create a microenvironment that promotes wound healing. M.R. El-Aassar et al. [73] fabricated electrospun nanofiber HA combined with polygalacturonic acid (PGA) and loaded it with AgNPs. Specifically, hydrophilicity and starin activity of the nanofiber was increased due to the increases in the HA content. The stress-strain curve of the (PGA/HA)-PVA nanofibers was much improved, and their capacity to withstand network deformation decreased as AgNPs concentration increased. This enhancement in nanofiber stress-strain was due to HA acting as a crosslinker between PGA/PVA chains. AgNPs presented in nanofibers had strong antibacterial effects against pathogens. In addition, AgNPs acted as antioxidant and anti-inflammatory agents and thus protected cells from the damaging effects of ROS and accelerated wound healing. Moreover, an in vivo evaluation in albino rats exhibited maximum epithelization and collagen deposition after 14 days in the presence of nanofiber.

Modified methacrylate hyaluronan (MHA) combined with polyacrylamide (PAAm) and loaded with AgNPs was used to prepare AHAs hydrogels (Q Tang et al. reported (2020) [74]. Engineered hydrogels exhibited satisfactory stretch, adhesion, and hemostatic characteristics. Rat-tail bleeding and liver bleeding models were used to investigate the hemostatic ability of the hydrogels. AgNPs containing hydrogels provide outstanding antibacterial activity against *E*. *coli* and *S. aureus*. AHAs hydrogels alleviated inflammation, promoted angiogenesis and collagen deposition, and improved granulation tissue development. Blood losses in the severed rat-tail and liver bleeding models showed that AHA hydrogels less blood clotting compare to the control. SEM images demonstrated the interaction between AHA or gauze and red blood cells many red cells adhered to the surface of AHA, presumably because of electrostatic interactions between positively charged MHA and negatively charged red blood cells, and the subsequent enhancement of platelet thrombosis. Zhang, Y et al. [75] constructed PCN-224-Ag-HA nanocomposite an antibacterial surface composed of a HA coating on photosensitive PCN-224 nanoscale metal-organic frameworks (nMOFs) embedded with AgNPs. Metal nodes of nMOFs selected as a photosensitive ligand 5,10,15,20-tetrakis(4-methoxycarbonylphenyl)porphyrin (TCPP) and Zr6 clusters. PCN-224-Ag-HA exhibited excellent biocompatibility with non-targeted bacteria and mammalian cells, which was attributed to the presence of HA and a small amount of Ag ion release. Moreover, PCN-224-Ag-HA was degraded and HAase was secreted in the presence of the bacteria to produce PCN- 224-Ag^+^, which electrostatically bound to bacteria. ROS might be exerted more efficiently as a result of the synergistic antibacterial action originating from Ag ions, allowing for resource conservation. PCN-224-Ag-HA demonstrated exceptional antibacterial activity in both in vitro and in vivo assessments, indicating that it has much promise for biomedical applications. Figure 4B displays the nano platform used for killing bacteria and disinfecting wounds using the Ag-infused nMOFs. Figure 4B(a) MRSA strain and Figure 4B(b) drug survival rates of *E. coli* under UV-light irradiation. (C, D) SEM images of MRSA and (D) Drug resistance for *E. coli* after various treatments under UV-light irradiation (1) PBS, (2) AgNO3, (3) PCN-224-HA, and (4) PCN-224-Ag-HA.

Tarusha, L. et al. [76] developed flexible HA-based alginate/Chitlac-AgNPs membranes and found they promoted wound healing in vitro and were effective against planktonic bacteria and bacterial biofilms. Furthermore, the Chitlac-AgNPs component was shown to suppress the proteolytic activity of matrix metalloproteinases (MMPs), which has been reported to hinder the healing of resistant wounds in vitro. Membranes were non-cytotoxic to keratinocytes and primary fibroblasts in vitro. Reswelling kinetic experiments showed this biomaterial is highly hygroscopic, which is critical for eliminating excessive exudates and removing bacteria nutrition from wound beds. In addition, membranes had a high water-vapor transfer rate (WVTR), which indicated they could maintain a moist environment at wound sites and prevent dehydration or exudate buildup.

### 4.2. Gold Nanoparticles

Gold nanoparticles (AuNPs) are now being employed to administer a variety of bioactive compounds and enhance therapeutic efficacy by providing tailored distributions, reducing toxicity, and increasing absorption [77,78], and are considered suitable transporters of proteins and macromolecules and the more widely distributed [79]. The use of AuNPs to treat various infections has been examined in various tissue damage models due to their antioxidant and anti-inflammatory properties [80]. Interestingly, Sumbayev et al. [81] showed that citrate-stabilized AuNPs reduced IL1-induced cellular response in vitro and in vivo. 

In an epithelial lesion Wistar rat model, Mendes, C. et al. [82] investigated the inflammatory and antioxidant effects of a combination of PBM and AuNPs-HA on proinflammatory IL1 and TNFα, anti-inflammatory IL10 and IL4, growth factors (FGF and TGFβ), and cytokines and on oxidative stress parameters. In addition, histological analysis was performed to evaluate inflammatory infiltrates, fibroblasts, new vessels, and collagen production. Treated animals had lower proinflammatory cytokine levels and higher anti-inflammatory cytokine levels. TGF and FGF levels also increased in treated animals, especially in the combination therapy group (PBM + AuNPs-HA). In terms of oxidative stress parameters, MPO, DCF, nitrite levels, and oxidative damage were lower in treated with carbonyl and thiol groups. Antioxidant defense was enhanced in the treated animals, and histologic sections showed inflammatory infiltration was lower in the PBM + GNPs-HA group than in the non-treated controls. PBM or PBM + HA treated animals had more fibroblasts, and collagen production was greater in all treated animals than in controls.

Ovais, M., et al. reported [83] Gold nanoparticles improve the healing of wound infection when used in photobiomodulation treatment. In addition, a cryopreserved fibroblast culture coupled with gold nanoparticles has been used to heal burns in rats. The antibacterial and antioxidant properties of gold nanoparticles have been shown to be extremely beneficial for wound healing and collagen tissue regeneration. In addition, AuNPs are involved in the release of proteins, such as IL-8, IL-12, VEGF, and TNF, which are important candidates for wound healing due to their antiangiogenic and anti-inflammatory activity.

### 4.3. Wound Healing Applications of HA-ZnO NPs

Metal oxides play a prominent role in the prevention of bacterial infections. The antimicrobial activity of ZnO NPs against microbial pathogens, such as Gram-positive and Gram-negative bacteria, is mediated by chemical processes such as the formation of ROS, the release of Zn^2+^ ions, and by physical effects, such as bacterial cell membrane rupture and internalization [84,85]. In addition, ZnO NPs promote tissue regeneration and vascularization, which aids wound healing [86,87]. ZnO loaded composite fibers exhibit antibacterial activity using two possible mechanisms: (i) The generation of reactive oxygen species, such as H_2_O_2_, O^2−^, and OH^−^, which causes oxidative stress in bacteria; and (ii) by interacting electrostatically with cell surfaces and damaging bacterial cell membranes [88,89,90]. When the effects of ZnO-loaded nanocomposite fibers were examined on *S. aureus* and *E. coli*, they were found to have greater antibacterial activity against *S. aureus*. This may have been because the cell walls of *E. coli* and *S. aureus* are quite different in terms of their structure and chemical compositions, that is, the cell wall of *S. aureus* is composed of a simple peptidoglycan layer, whereas that of *E. coli* contains lipid A, lipopolysaccharide, and peptidoglycan [91,92]. Furthermore, the outer layer of the *S. aureus* bacteria might increase ZnO adherence, whereas that of *E. coli* may reduce attachment [93].

The antimicrobial activity of ZnO NPs is due to the disruption of bacterial cell membranes and the stimulation of reactive oxygen species (ROS) [94]. Different types of biomaterials with antimicrobial properties produced by embedding ZnO NPs into hydrogel networks have been used to promote wound healing. However, most of these materials have low adhesion strengths and are not easily biodegraded [95,96]. Thus, there is a need to develop composites with better adhesive and mechanical strengths and biodegradation properties to encourage skin repair and regeneration [88]. Table 3 shows summarize MO NPs-loaded HA composites used for wound healing applications. 

Non-antibiotic combined treatments based on functionalized nanofibers (NFs) have been used to suppress microbial invasion and minimize antibacterial resistance [97]. In a new approach to the functionalization of NFs for wound healing, a non-antibiotic combinational therapy was employed to limit microbial invasion and decrease antimicrobial resistance. M.R. El-Aassar et al. [98] produced NFs by embedding a ZnO NPs/cinnamon essential oil (CEO) as a biocomposite in an electrospinnable HA/polyvinyl alcohol (PVA)/polyethylene oxide (PEO) blend. The effect of this material on bacterial growth was performed using log number plots of S. aureus [99]. HA/ZnO/CEO-NFs reduced *S. aureus* growth more than HA/ZnO-NFs or HA/CEO-NFs, presumably due to the presence of ZnO NPs [100] and CEO [101]. NFs are non-toxic to human dermal fibroblasts (HDF) cells [102]. Furthermore, it has also been reported that HA-ZnO nanocomposite enhanced the cytocompatibility of essential oils [103,104]. After 24 h, all antibacterial NFs formulations showed excellent antibacterial activity against *S. aureus*. The lower in vivo activity of singly loaded NFs than combinational NFs was attributed to more challenging wound microenvironments [105], prolonged interactions with wounds, and continuous CEO release. As a result, the combinational NFs displayed antibacterial properties. Hadisi, Z et al. [106] fabricated a wound dressing using core-shell hyaluronic acid–silk fibroin/ZnO nanofibers to treat burn injuries. ZnO NPs incorporated in the core of nanofibers improved sustained drug release and regulated bioactivity. Increases in the ZnO content in the polymer matrix increased wound dressing antibacterial abilities against *E. coli* and *S. aureus*. When HA–SF nanofibers were tested *in vitro*, HaCat cells displayed good cell adhesion, cell proliferation, and viability in culture at ZnO doping levels up to 3%. Since HA–SF/ZnO-3 stimulated HaCat cell movement, a similar tendency was found in scratch experiments. The inclusion of ZnO may have increased wound contraction in the HA–SF/ZnO-3 group. ZnO has also been demonstrated to have favorable effects on the stimulation and migration of epithelial cells and keratinocytes and wound closure [107,108]. Based on the outcomes of in vitro experiments, HA–SF/ZnO-3 was chosen for an in vivo examination. Histologically, HA–SF/ZnO-3 improved burn wound healing and skin regeneration and enhanced collagen deposition by stimulating the creation of epidermis, hair follicles, and sebaceous glands. Furthermore, immunohistopathological staining revealed that the HA–SF/ZnO-3 nanofiber matrix-treated burn site reduced inflammatory response than gauze or HA–SF. Figure 5A depicts the fabrication of antibacterial NFs with a core-shell structure for wound dressing. The majority of studies on ZnO NPs in hydrogels have reported ZnO NPs suitable for wound dressings. in-situ production of ZnO NPs has been employed in a few studies [109]. Rao et al. [110] reported the simple in-situ formation of ZnO NP nanobelt-like structures from HA hydrogel crosslinked with 1, 4-butanediol diglycidyl ether (BDDE) Figure 5B. HA-ZnONPs may be superior candidates for wound healing applications due to their cell-adhesive, hemostatic, and antibacterial activities. The nano belt-like structure (Figure 5C) plays a prominent role in the swelling and biodegradation of HA hydrogels. HA-ZnONPs exhibited enhanced blood clotting and hemocompatibility, which was as good as that of antibacterial agents (Figure 5D). The hydrophilicity of HA polymeric matrices, which reduce RBC destruction, might be due to lower hemolysis by hydrogels. Furthermore, HA hydrogels with or without ZnO nanobelt-like features have high hemocompatibility and percentage hemolysis values of < 5%. In addition, CCD-98sk cells treated with HA-ZnO NPs show good adhesion and proliferation. Consequently, ZnO nanobelt-like structures did not affect the cell adhesion properties of CCD-986sk cells. Moreover, HA plays a vital role in the migration of fibroblasts, which produce ECM for wound healing [111,112]. 

The use of MNPs and MO NPs combinations to reduce cell damage and enhance antibacterial efficacy is an active research topic. HA-based on AgNPs and graphene oxide (GO), Ran, X et al. [113] developed a hyaluronidase-triggered photothermal antibacterial platform. The antibacterial activity of HAase-triggered release was outstanding against *S. aureus*. GO-based nanomaterials elevated temperature locally when exposed to NIR light and induced substantial bacterial mortality. The antibacterial Haase-triggered AgNPs release technique provides that the HA template protects AgNPs, while causing no harm to mammalian cells. The nanocomposites exhibited antibacterial action against *S. aureus* and were non-toxic to mammalian cells. Furthermore, HA-GO-AgNPs demonstrated outstanding antibacterial activities in a wound disinfection model in vivo. 

**Table 3 pharmaceutics-14-02235-t003:** HA-MO NPs for wound healing applications.

Base Polymer	Other Polymer	Nanoparticles	Cross-Linker	Type of Composite	Technique	Bacterial Strains	Cells	Ref.
HA	PVA/PEO	ZnO	Glutaraldehyde S(GA)	Nanofiber	Electrospinning	*S. aureus*	HDF cells	[98]
HA	Silk Fibroin	ZnO	--	Nanofiber	Electrospinning	*E. coli* and *S. aureus*	HaCat cells	[106]
HA	--	ZnO	1,4-butanediol diglycidyl ether (BDDE) cross-linker	Nano-belt like structure	in-situ free-radical polymerization	*E. coli* and *S. aureus*	CCD-98sk cell	[110]

## 5. Drug Loaded HA Nanocomposites for Wound Healing Applications

The wound healing process can be accelerated by HA alone but this must be done in different ways. Sustained release of a therapeutic payload can provide more effective treatment. Although many drugs can be used as wound healing treatments, the inflammatory environments of wounds make it difficult for a drug to promote healing, and few candidates have proven clinical outcomes. HA and its compounds help to deliver antibacterial agents, balance inflammation at wound sites, and enhance the healing process. Regarding recent advancements, the development of eco-friendly, cost-effective, multi-functional, and transparent wound care dressings with excellent exudate interactions and wound adherences, while encouraging healing and preventing bacterial biofilm development and infections present an intriguing challenge. Furthermore, constructing such structures with energy-efficient techniques and environmentally friendly materials and solvents (i.e., aqueous) would be a significant advantage. Table 4 shows summarize drug-loaded HA-nanocomposites used for wound healing applications. 

M. Contardietal et al. [114] developed a multifunctional bi-layer wound dressing with a HA/polyvinylpyrrolidone (PVP) transparent matrix containing and antibiotics, respectively. The bilayer structure consisted of a PVP top sheet containing the antibiotic Neomercurocromo^®^ (Neo) (Pavia, Italy) and a HA/PVP bottom sheet containing the antibiotic ciprofloxacin (Cipro). This bi-layer system is transparent, self-adhesive, flexible, and was found to be biocompatible in vitro when tested on human foreskin fibroblasts. In addition, skin disinfectants and antibiotics in this system allowed drug release in a controlled manner over 5 days. Three distinct strains of *S. aureus*, *E. coli*, and *P. aeruginosa* were tested for antibacterial activity. Furthermore, the construct’s capacity to be entirely resorbed by wounds was also demonstrated in an in vivo full-thickness excision model; it was probably integrated into healed tissues. HA is commonly dissolved in toxic organic solvents to synthesize electrospun nanofibers. Although HA is water-soluble, its ionic nature causes long-range electrostatic interactions, and the presence of counter ions causes dramatic increases in the viscosities of aqueous HA solutions without ensuring sufficient chain entanglement for stable and effective electrospinning. Although electrospinning of HA containing PVA as a carrier polymer permitted processing in water, the scaffolds were inefficient and had several defects.

M. Seon Lutzetal et al. [115] proposed an HA-based nanofiber scaffold containing hydroxypropyl β-cyclodextrin (HPβCD) using water as a solvent for safe and functional wound dressings. Poly (vinyl alcohol) (PVA) is used as a carrier polymer, and HPβCD stabilizes the electrospinning process and enables the formation of uniform nanofibers. in-situ crosslinking processes for scaffolds have also been proposed to ensure products are non-toxic. Furthermore, the inclusion of HPβCD in HA fibrous scaffolds paves the way for the production of wound dressings with regulated drug release and encapsulation abilities. The scaffold was impregnated with the non-steroidal anti-inflammatory drug naproxen (NAP) in an aqueous solution or supercritical CO_2_. The functional scaffolds produced had consistent drug kinetic release profiles and retained their fibrous integrities over several days. 

A Cur-HA-Spu (Cur = Curcumin and Spu = Succinylated pullulan) composite production by esterification was reported by Y. Duan et al. [116]. Cur-HA-Spu demonstrated a high swelling ratio, rapid hemostasis ability, antimicrobial activity, and antioxidant characteristics. MTT and proliferation tests in L929 cells demonstrated that Cur-HA-Spu polymer was non-cytotoxicity and increased cell proliferation as compared with Cur. Cur-HA-Spu exhibited antibacterial activity against *E. coli* and *S. aureus*. The materials produced also demonstrated antioxidant activity when tested using the DPPH technique. Cur-HA-Spu film produced superior wound healing results than HASPu film and spontaneous healing in a Wistar rat investigation. Cur has various advantages for wound healing, that is, it has antioxidant, anti-infectious, and anti-inflammatory effects [117,118] and encourages granulation tissue formation, collagen deposition, tissue remodeling, and wound contraction, but the effectiveness of Cur is limited by its poor solubility (≤0.125 μg/mL). However, grafting with HA to produce Cur-HA-Spu increased the solubility of Cru to 3.1∼5.3 μg/mg. Cur can speed up the different stages of wound healing, including inflammation, proliferation, and maturation. Furthermore, it can also efficiently scavenge ROS and promote antioxidant enzyme synthesis in wound environments during the inflammatory phase. In the proliferation phase, Cur also promotes fibroblast migration to form granulation tissue and re-epithelialization, and in the maturation phase, it increases cytokine levels in wounds to facilitate wound contraction and fibroblast proliferation [119]. Nanoparticle inclusion, double-network, and double-crosslinking techniques have all been utilized to increase the mechanical strengths of hydrogels [120]. 

Haopeng Si et al. [121] prepared a UV-crosslinked methacrylic anhydride (HA-MA) combined with HA-SH/3, 3′-dithiobis (propionyl hydrazide) (DTP) crosslinked network hydrogel bioink for 3D bioprinting. Different weight ratios of HA-MA and HA-SH changed the rheological properties of the hydrogels; for example, HA-MA concentration enhanced the storage modulus (G’) of the hydrogel matrix. Moreover, hydrogels had a higher swelling ratio and exhibited controlled degradation. When nafcillin was incorporated into hydrogels, it exhibited good antibacterial activity. In addition, rheological and swelling studies, drug release kinetics in vitro degradation, and cytotoxicity characteristics of nafcillin-loaded hydrogel were examined. HDF cells were used for in vitro cytotoxicity assays of nafcillin-loaded HA-MA/HA-SH hydrogels. Figure 6 provides a summary of drug release at wound sites, bacteria mortalities, and regeneration skin effects of HA combined with different drugs. Table 3 summarizes drug-loaded HA-nanocomposites used for wound healing applications. 

### HA Combined with Antibiotics and MNPs

The demand for novel antimicrobial strategies is being driven by increased severe infections caused by antibiotic drug resistance and a fall in the number of new antibacterial medications licensed for usage. N.Yu, et al. [122] adevised a HA/AgNPs/gentamicin nanocarrier (HA/Ag/g) as a multi-responsive antibacterial nanocarrier and combined it with mussel-inspired chitin hydrogel. A simple self-assembly process was used to prepare HA/Ag/g, which exhibited controlled drug release; Ag was stimulated by bacteria-produced Haase or pH. HA/Ag/g@CPH inhibited bacterial growth and had good adhesion but no effect on cell adhesion or proliferation. In vivo studies revealed that HA/Ag/g@CPH might promote wound healing. The rapid evolution of drug-resistant bacteria and the significant delay in the creation of new medicines underlie the need for antibacterial research. Nanomaterials with different effect sizes and antibacterial properties might be useful alternatives to antibiotics. For this reason, the chemo-photothermal enzyme-responsive drug delivery nanosystem was developed by Liu, Y., et al. [123]. The AA@Ru@ HA-MoS_2_ nanosystem used was composed of mesoporous RuNPs encapsulated by ascorbic acid (AA)-capped HA. Ciprofloxacin (CIP) coated (molybdenum disulfide) MoS_2_ nanoparticles inhibited the bacterial activities of *P. aeruginosa* and *S. aureus* and efficiently accumulated at wound sites, and the HA capping agent was dissolved by bacteria to produce Hyal at sites of infection. Subsequently, the encapsulated AA was catalytically decomposed by MoS_2_ to form hydroxyl radicals (•OH). AA@Ru@ HA-MoS_2_ exhibited good NIR photothermal response and synergetic antibacterial activity. The fabricated nanosystem shown in Figure 7A induced bacterial morphologies changes. Bacterial cell membranes were oxidatively damaged by converting AA to •OH at sites of infection. •OH specific release under NIR exposure destroyed bacterial cell membranes and caused the leakage of bacterial contents. This result was attributed to the inclusion of CIP in the AA@Ru@HA-MoS_2_. 

Y. Liang et al.,[124] prepared HA grafted dopamine (DA) loaded with reduced graphene oxide (rGO) using H_2_O_2_/HPR (horseradish peroxidase) as an antibacterial, antioxidant, photothermal hydrogels for wound dressing applications (Figure 7B). Fabricated hydrogels were self-healing due to hydrogen bonding and π-π- stacking occurs between rGO@PDA and HA-DA, and hydrogel stability, gelling, crosslinking, mechanical properties, and conductivity improved when the rGO@PDA content was increased. Furthermore, the inhibition zones for *S. aureus* and *E. coli* increased for HA-DA/rGO hydrogels when the irradiation period was increased from 0 to 10 min and showed potent photothermal bacterial activity in vitro and in vivo. When compared with commercial wound dressing films (Tegaderm™), doxycycline-loaded hydrogels (HA-DA/rGO3/Doxy) greatly promoted wound healing.

## 6. HA-Derivatives and Other Polymers Used for Wound Healing Application

Nanoparticles have been employed as novel wound healing therapies. AgNPs act as anti-infection agents and AuNPs stimulate fibroblast proliferation and collagen production [125]. However, AgNPs and AuNPs may accumulate subdermally and are difficult to excrete [126]. Furthermore, the cost of these materials presents a substantial barrier to their widespread clinical and outpatient use. Figure 8 shows summarize HA and other polymer structure and significant properties in wound healing.

Li X. et al. [127] developed a HA/poloxamer (HA-POL) hydrogel and investigated its therapeutic efficacy on skin wound healing. HA-POL solution became a gel at 30 °C and retained its moisturizing properties. Furthermore, the air permeability of HA-POL hydrogel was greater than that of a conventional wound dressing. According to the results of Transwell experiments, the HA-POL hydrogel successfully prevented bacteria (*E. coli*) invading skin wounds. Moreover, HA-POL hydrogel promoted fibroblast growth factor (bFGF) production and thus wound healing. In addition, hydrogels promote wound healing by increasing protein deposition in wound sites. 

Uddin et al. [128] constructed two assemblies of polyelectrolyte multilayers (PEMs) using a layer-by-layer technique at three different insoluble multi-l-arginyl-poly-l-aspartate (iMAPA) with HA (iMAPA/HA) to γ-polyglutamic acid (iMAPA/γ-PGA) concentration ratios. The effects of iMAPA and its counterparts, HA or γ-PGA, as a terminal layer on film roughness, cell proliferation, and cell migration were investigated. Because of stronger charge interactions, iMAPA incorporation was greater at high anionic polymer concentrations, and iMAPA/HA films were smoother than iMAPA/γ-PGA multilayers. L929 fibroblast growth rates on PEMs were comparable to those on a glass substrate, with no additional benefit from the terminal layer. However, all PEMs boosted the migratory rates of L929 cells as compared with untreated glass, γ-PGA integrated films promoted cell migration by 50% after 12 h of culture, while smooth films containing HA increased cell migration by up to 82%. The results obtained showed the use of iMAPA to create layer-by-layer systems of polyelectrolyte biopolymers might have application in wound dressings.

Liu S et al. reported [129] bioadhesive hydrogels prepared by combining oxidized hyaluronic acid (HA-CHO) and dopa-grafted ε-polylysine (EPL- Dopa). ‘An enzymic Schiff base crosslinking reaction was used to fabricate HA/EPL hydrogel dressings. Bioadhesive hydrogel has inherent antibacterial properties due to its high positive surface charge density and may efficiently kill *E. coli* and *S. aureus* bacteria. The physicochemical characteristics of HA/EPL were investigated in vitro and included gelation time, internal structure, swelling and degradation behavior, and rheological, self-healing, and adhesion capabilities. The cytocompatibilities of hydrogel dressings and prepolymer solutions were determined in vitro using L929 cells, the Cell Counting Kit-8 (CCK-8) technique, and live/dead assays.

P. Luo et al. [130] studied adipic acid dihydrazide modified hyaluronic acid (ADH-HA) combined with oxidized hydroxyethyl cellulose (OHEC). The resulting hydrogel had a shortest gelation time of < 106 sec and a maximum swelling rate of 2888%, though the swelling rate decreased as OHEC oxidation increased. Hemolysis tests revealed that the hydrogel was compatible with blood; hemolysis rates ranged from 2.4% to 4.0%. Furthermore, cytotoxicity studies revealed that the hydrogel was not toxic to NIH-3T3 cells and that all samples increased cell viability by 85%. The remarkable performance of this hydrogel suggests its use as a wound dressing.

H. Ying and colleagues [131] produced a collagen/HA (COL-HA) hydrogel by connecting the phenol moieties of collagen I-hydroxybenzoic acid (COL-P) and hyaluronic-acid-tyramine (HA-Tyr) in situ using horseradish peroxidase (HRP). The hydrogel produced had strong antibacterial effects on *E. coli* and *S. aureus*. The porous nature of the COL-HA hydrogel permitted the interchange of gas, medium, and nutrients, and COL-HA substantially promoted the proliferation of human microvascular endothelial cells (HMECs) and fibroblasts (COS-7). More crucially, a vascular endothelial growth factor (VEGF) was detected in the HMEC grown hydrogel, which suggested that vascular regeneration might be possible. Because COL and HA combine to promote wound repair, the healing ratio, and efficacy of full-thickness wounds treated with COL-HA hydrogel were greater than those treated with commercial drugs, COL-P hydrogel or HA-Tyr hydrogel. 

L Hong et al. [132] reported the preparation of two classes of HA-based hydrogels by freezing-thawing (HA1) or chemical crosslinking (HA2). Using New Zealand rabbits and powdered HA and cotton clothing as references, both hydrogels were applied to cure full-thickness skin lesions. After disinfecting wounds with iodine and treating them with HA2, HA1, HA, or cotton dressing (the control) the wounds begin to heal. Healing progress was monitored and evaluated over 56 days, and the biological mending mechanism was investigated. Based on wound area changes, white blood cell (WBC) counts, and H&E staining results, HA2 was the most promising therapy in terms of encouraging wound healing with minimal scar formation. Immunochemistry and real-time PCR of the bio-factors involved in wound healing, that is, vascular endothelial growth factor (VEGF), alpha-smooth muscle actin (α-SMA), and transforming growth factor beta-1 (TGF-1), revealed that HA2 increased VEGF and α-SMA secretion but decreased TGF-1 expression at an early stage. Furthermore, HA2 reduced wound inflammation and scar formation and improved skin regeneration. 

A. Eskandarinia, et al. [133] used a solvent-casting method to prepare CS/HA/EEP films from a combination of HA, cornstarch (CS), and the ethanolic extract of propolis (EEP). Films were characterized for molecular interactions and surface morphology, assessed for opacity, EEP release, and equilibrium swelling, and subjected to in vitro and in vivo evaluations. CS/HA/EEP-0.5% film dressings had better antibacterial activities against *S. aureus*, *E. coli*, and *S. epidermidis* than CS, CS/HA, and CS/HA/EEP-0.25% films. In addition, CS/HA/EEP-0.5% had no toxic effect on L929 fibroblasts. Furthermore, the wound healing process might have been accelerated in skin excisions by CS/HA/EEP wound dressings in Wistar rats. These results suggest that adding HA and EEP to cornstarch wound dressings can greatly improve wound healing efficacy. 

The potential of *N*-butylated LMW-HA (BHA) for skin healing both in vitro and in vivo was examined by Yin Gao et al. [134]. BHA was found to improve skin healing significantly more than a commercial wound dressing material. Wound closures achieved by partially de-acetylated LMW-HA (DHA) and re-acetylated DHA (AHA) were significantly postponed, which demonstrated the importance of the N-acylation of LMW-HA. In a systematic study, these authors showed that the therapeutic effects of BHA were achieved by targeting inflammation, proliferation, and maturation. Supramolecular hydrogels (SH) provide reversible, dynamic, and biomimetic control over structural characteristics. For therapeutic applications, the development of SH using enhanced structural and functional recapitulations of injured organs is critical. For this reason, Weiyi Zhao et al. [135] prepared a photo-responsive SH via host-guest interactions between azobenzene and β-cyclodextrin groups attached to HA chains. SH with a dynamic spatial network crosslink density was created by applying a light stimulus using the photoisomerization characteristics of azobenzene at different wavelengths. It was suggested that the loosened hydrogel might rapidly release EGF when exposed to UV light and enhance EGF delivery at the wound site. Controlled EGF release from a supramolecular hydrogel displayed higher wound healing effectiveness in terms of granulation tissue development, growth factor levels, and angiogenesis, according to an in vivo assessment of the healing process using a full-thickness skin defect model. Table 5 details HA/polymer combinations used cells for wound healing applications. 

## 7. Clinical Perspectives

Although some of HA-based products available in the market [136], There are still challenges to be addressed the HA-based nanocomposites for clinical study in animal models. Few investigations have been applied clinical studies in animal models using the HA nanocomposites Ag or Zno, or GO/Ag, Ag-nMOFs as well as MNPs with antibitic drugs, such as ciprofloxacin and doxycycline, etc., for wound healing properties by in vivo. The results proved the effective healing of wound bacterial infected and diabetic wounds. The use of antibacterial agents improves the wound healing property as explained in Figure 2 (mechanism action of antibacterial activity of HA/nanocomposites). Several researchers have used Ag nanoparticles embedded in HA-based composites for wound healing by in vivo animal models. They have shown great potential for improving the wound healing property [69,71,74]. The combination of GO/Ag further improved the wound healing of bacterial wounds due to its NIR active property, which produced thermal heating to kill bacteria [105]. The combination of the antibiotic drug with metallic nanoparticles also proved the effective inhibition of bacterial growth to improve wound healing by in vivo [122]. The combination of rGO@PDA-HA_DA hydrogels proved the effective inhibition for *S. aureus* and *E. coli* bacterial via photothermal effect as compared to a commercialized wound dressing films (Tegaderm^TM^) [124]. Although HA-based nanocomposite has shown improved abilities to aid in the healing of infected wounds, practical use is still difficult because of associated impurities during synthesis, which should be carefully controlled to avoid excessive inflammatory reactions. Additionally, the sterilization of products based on untreated or modified HA may also degrade the polymer, affecting its immunological reactions. Advanced HA-based solutions for use as wound dressings, however, encounter a series of technical challenges during product development, including good mechanical qualities, outstanding biological results, and affordable production costs without the use of harmful ingredients. However, the in vitro and in vivo experiments of HA-based scaffolds that have been reported by a number of researchers show that HA is a promising material and that these nanocomposites with remarkable therapeutic properties may be taken to use in the clinical purpose in future. 

## 8. Future Perspectives and Challenges

Presently very limited exploration for control and acceleration to wound healing by using HA-based nanocomposite. Specifically, HA-based metal (or) metal-oxide, such as Ag, Au, Ru, and ZnO, are used for controlling and killing bacterial pathogens in the wound healing process. In addition, antibiotic drugs and combinations with metal/metal oxides as well as other suitable polymers are used as wound dressing materials. Furthermore, antibiotics are the most widely used drugs in the clinical setting, and controlled-release antibiotics have attracted considerable interest. However, antibiotic abuse still exists and there is no unified standard for evaluating the controlled release of these drugs to ensure optimal therapy efficacy. However, bacterial killing by using the HA-nanocomposite mechanism difficult to understand to control and promote wound healing in vitro and in vivo process. Therefore extended an understanding of the mechanism for different bacterial agents including Cu, Ce_2_O, MgO, etc. for control and promoting the wound healing process. In addition, HA-modified or quaternary amine polymers exhibited excellent antibacterial properties for control and accelerating wound healing. Wound healing is complex and can be improved using functional dressings that release drugs or growth factors. Accordingly, HA-based nanocomposite wound dressings have been produced that release antibacterial agents or other active agents in a sustained manner. It is conceivable that some of the physical and chemical features of biological tissues could be mimicked by new bioactive nanomaterials, such as clay, ceramics, or metallics, and that these bioactive materials could control infections and enhance wound healing. We believe further manufacturing and multicomponent system developments will result in the next generation of wound-healing nanomaterials. 

## 9. Conclusions

A variety of engineered nanomaterials have been produced with the joint aims of regulating wound infections and accelerating wound healing. HA is a unique material in terms of achieving these objectives due to its biocompatible and biodegradable properties and its limited adhesion to bacteria. In addition, it can be altered in various ways to enhance its properties. This review mainly described HA’s important role in wound healing’s four stages and it explained engineered HA nanocomposite scaffolds containing MNPs or MO-NPs, antibiotics and other polymers to control wound infection and promote healing. Furthermore, clinical perspectives of HA-based nanocomposites used for wound healing applications using in vivo methods. In particular, MNPs, MO-NPs and antibiotics can inactivate bacterial pathogens by damaging the cell wall of the bacterial membrane, transferring electron chains, nucleic acids, proteins or enzymes. This damage can be generated by direct (adhering to biological molecules) or indirect (generating ROS) mechanisms. Accordingly, HA-based nanocomposite wound dressings were produced that sustainably release antibacterial agents, such as Ag, Au, Ru or ZnO, GO, growth factors, or other active agents. Furthermore, antibiotic drugs alone or in combination with MNPs/MO-NPs effectively kill the bacteria in a sustained manner for wound healing applications. Additionally, the inherent antibacterial killing properties of polymers are used directly or to optimize the polymer in combination with the HA composite for wound healing applications. HA-based nanocomposite materials accelerate the wound healing process in vitro and in vivo.

## Figures and Tables

**Figure 1 pharmaceutics-14-02235-f001:**
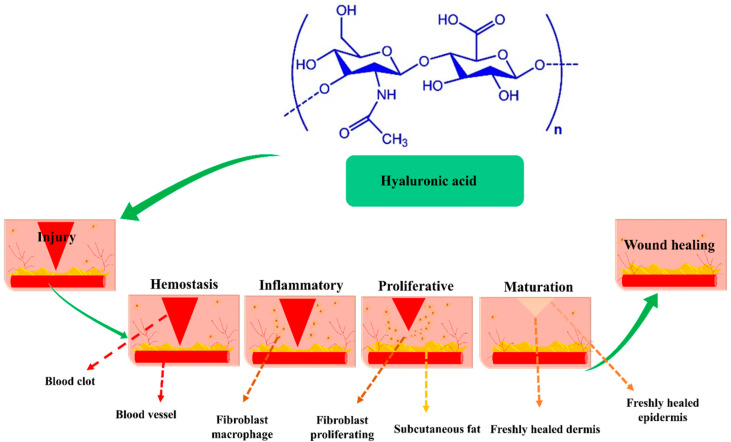
The influence of HA at wound sites during the hemostasis, inflammation, proliferation, and remodeling phases of wound healing, Copyright 2017, ACS publication [36].

**Figure 2 pharmaceutics-14-02235-f002:**
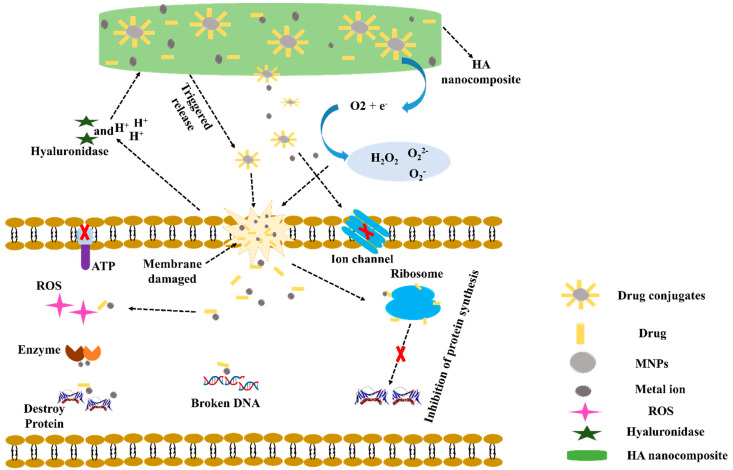
Possible antibacterial mechanism for a metal−based HA composite drug Copyright 2017, Elsevier [64].

**Figure 3 pharmaceutics-14-02235-f003:**
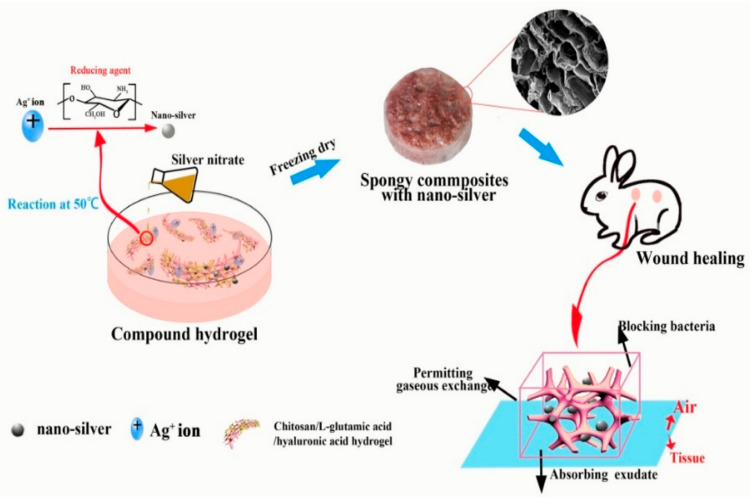
Summary of the AgNPs formation to words bacterial killing for wound healing. Copyright 2017, Elsevier [70].

**Figure 4 pharmaceutics-14-02235-f004:**
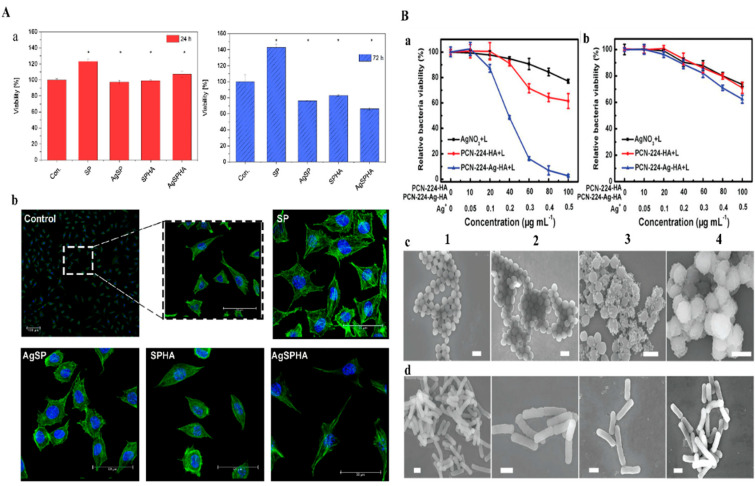
**A**(**a**) Cytotoxicity of at 24 and 72 h **A**(**b**) L929 Cell morphologies control and thermosensitive hydrogels for 1 day. DAPI stained nuclei (blue) and actin filaments stained with phalloidin−TRIC (green). Copyright 2019, Elsevier [71]. **B**(**a**) MRSA strain and **B**(**b**) drug resistance survival rates against *E. coli* under UV-light irradiation. SEM images for (**c**) MRSA and (**d**) drug resistance for *E. coli* after various treatments under UV-light irradiation (1) PBS (2) AgNO_3_ (3) PCN−224−HA and (4) PCN-224−Ag−HA. Copyright 2019, John Wiley and Sons [75].

**Figure 5 pharmaceutics-14-02235-f005:**
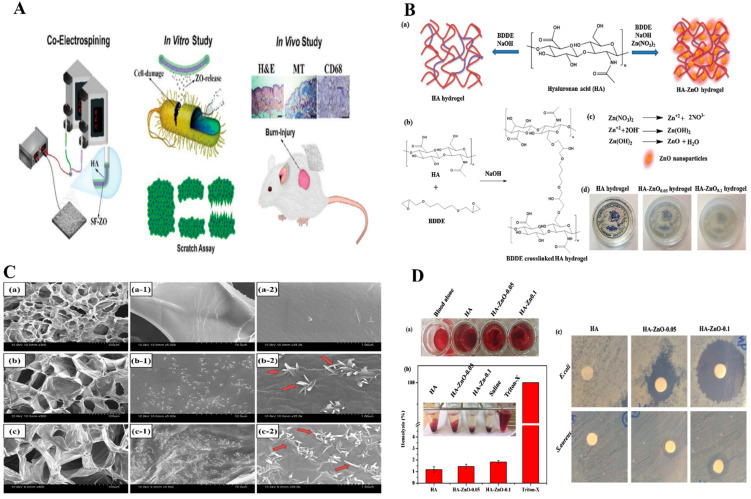
(**A**) Preparation of antibacterial HA-SF/ZnO fibers with core-shell structure and applied for wound healing. Copy right 2020, John Wiley and Sons [106]. (**B**) Formation of HA and HA-ZnO hydrogels, (**C**). SEM images ZnO hydrogels. (**D**). Hemolysis and antibacterial activity of hydrogels Figure (**B**–**D**) Copyright 2019, Elsevier [113]. (**B**) HA and HA-ZnO hydrogels **B**(**a**) formation of mechanism **B**(**b**) BDDE crosslinked hydrogels, **B**(**c**) ZnO nanoparticles, and (**d**) digital photographic hydrogels. (**C**) SEM images of **C**(**a**–**a-2**) HA, **C**(**b**–**b-2**) HA-ZnO-0.05, **C**(**c**–**c-2**) HA-ZnO-0.1. (**D**) Hydrogels treated with pig whole blood, **D**(**a**) photographic images of blood clot formed, **D**(**b**) hemolysis (%) of different hydrogels (inset photographic images showing hemolysis), **D**(**c**) antibacterial activity of hydrogels against *E. coli* and *S. aureus*

**Figure 6 pharmaceutics-14-02235-f006:**
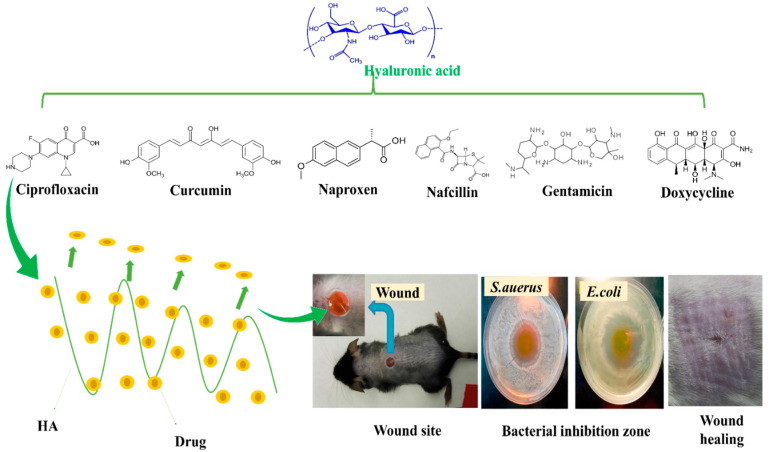
HA combined with different drugs; release at wound sites, bacteria mortality, and skin regeneration. Copyright 2017, Elsevier [116].

**Figure 7 pharmaceutics-14-02235-f007:**
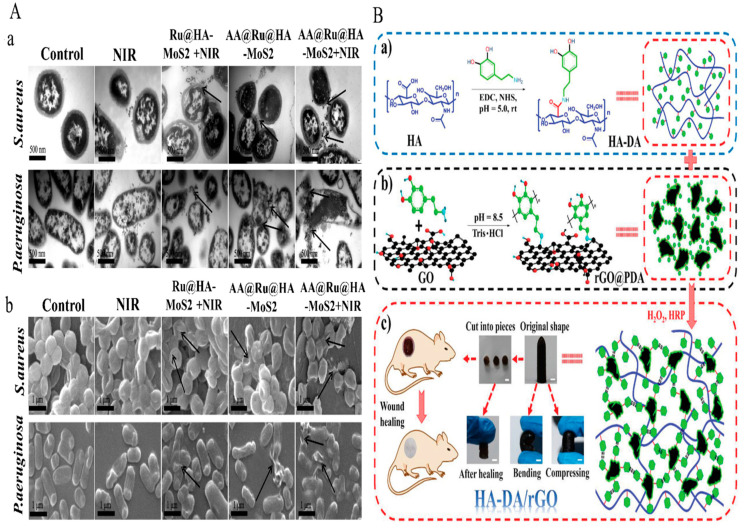
(**A**) TEM and SEM images of MDR bacteria treated with HA-nanosystem. Copy right 2019, American Chemical Society [123]. (**B**) (**a**) Preparation of the HA-DA/rGO hydrogel (**b**) HA-DA polymer formation for rGO@PDA, (**c**) HA-DA/rGO self-healable hydrogel applied in wound healing applications. Copy right 2019, John Wiley and Sons [124]. (**A**) TEM (**a**) and SEM (**b**) images of MDR bacteria treated with NIR, Ru@HA-MoS2 + NIR, AA@Ru@HA-MoS2, and AA@Ru@HA-MoS2 + NIR for 2 h respectively [123].

**Figure 8 pharmaceutics-14-02235-f008:**
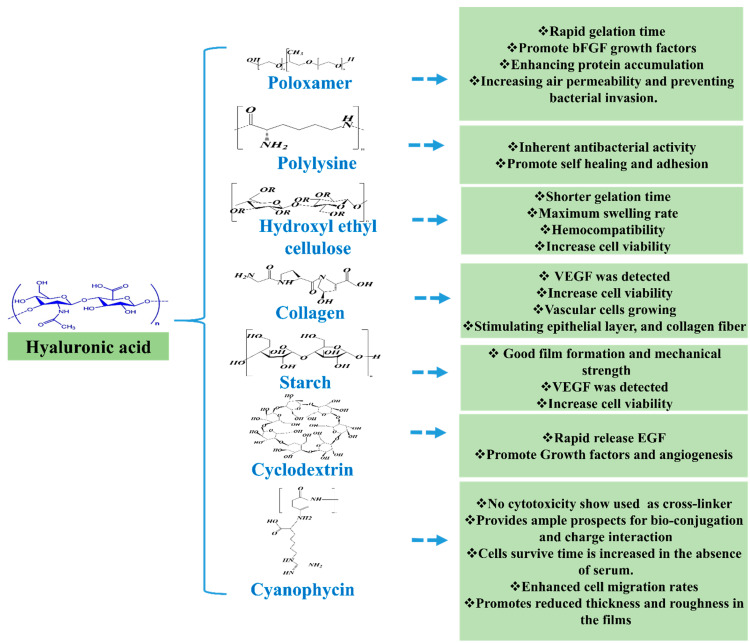
Summarizes HA/polymers structure and significant properties in the wound healing process.

**Table 1 pharmaceutics-14-02235-t001:** A Summary of Wound Healing.

Activity	Hemostasis	Inflammation	Proliferation	Remodeling
Healing Activity	Hours Vasoconstriction Platelet AggregationPlatelet DegranulationBlood clotting	Days Leucocyte migration Neutrophil activationsKilling bacteria Excluded cellar derbies Liberation of Growth factors	Week Neo-vascularisation Angiogenesis Fibroblast proliferation Kertinocytes migrations Collagen type III formation	Months Fibroblast secretion ECM reorganization Collagen type I formation
Cell type	Platelet	Neutrophil	Keratinocyte, Endothelial, Fibroblast,	Fibroblast
Cytokine and Growth factors	Fibrin Thrombin Clotting factor I	TLR2, TLR4 TGF-β, VEGF TNF-α	TNF-α, VEGF bFGF, CD44	MMPs TGF-β

**Table 2 pharmaceutics-14-02235-t002:** HA-based Ag nanoparticles for wound healing applications.

Base Polymer	Other Polymer	Nanoparticles	Cross-Linker	Type of Composite	Method	Bacterial Strains	Cells	Ref.
HA	----	AgNPs (20–25 ± 2 nm)	----	Fabric nanocomposite	The wet dry-spinning technique (WDST)	*E. coli*	Keratinocyte cell line (*HaCaT*)	[69]
HA	Chitosan-L-glutamic acid (CG)	AgNPs (5–20 nm)	----	Spongy composite	Freeze-drying	*E. coli* & *S. aureus.*	L929 cells	[70]
HA	corn silk extract (CSE)	AgNPs (13 ± 1 nm)	self-assembling	Hydrogels	microwave-assisted	*B.subtilis*, *S. aureus* & *P. aeruginosa*, *E. coli*	L929 cells & HDF cells.	[71]
HA	Polygalacturonic acid (PGA)	AgNPs	----	Nanofibers	Electrospinning	*B.subtilis*, *S. aureus* & *E. coli*	----	[73]
HA	Gelatin	AgNPs	(EDC NHS) Photo cross-linker	Hydrogels	Free-radical polymerization under UV light	*S. aureus* and *E. coli*,	3T3 cells	[74]
HA	PCN-224	AgNPs	Visible lightTCPP	Hydrogels	Photosensitive	MRSA *E. coli*	L929 cells	[75]
HA	Alginate with chital -AgNPs	AgNPs	Calcium ion	Spongy membrane	Freeze-drying	*S. aureus*, *S.epidermid* & *P. aeruginosa*	HaCaT & HDF cells	[76]

**Table 4 pharmaceutics-14-02235-t004:** HA-nanocomposite loaded with antibiotic drugs for wound healing applications.

Base Polymer	Other Polymer	Nanoparticles	Composite Structure	Drug	Bacterial Strains	Cells	Ref.
HA	polyvinylpyrrolidone (PVP)	----	Bilayer films	Ciprofloxacin (Cipro)	*S. aureus*	HDF cells	[106]
HA	poly(vinyl alcohol) (PVA)	----	Nanofiber	Naproxen (NAP)	*S. aureus, E. coli & P. aeruginosa.*	HaCat cells	[107]
HA	Spu	----	Films	Curcumin	*S. aureus,& E. coli*	L929 cells	[108]
SHA	----	----	Hydrogels	Nafcillin	----	HDF cells	[113]
HA	----	AgNPs	Hydrogels	gentamicin	----	----	[114]
HA	AA	Ru	Nanocomposite	ciprofloxacin (CIP)	*S. aureus & P. aeruginosa*	----	[115]
HA	DP	DP-rGO	Hydrogels	doxycycline	*E. coli* and *S. aureus*	L929 cells	[116]

**Table 5 pharmaceutics-14-02235-t005:** HA combined with other polymers for wound healing applications.

Base Polymer	Polymer Combination	Cross-Linker	Composite Structure	Technique	Bacterial Strains	Cells	Ref.
HA	Poloxamer	----	Hydrogels	Sol-gel	*E. coli*	Fibroblast	[127]
HA	Multi-l-arginyl-poly-l-aspartate (MAPA) and γ-polyglutamic acid (γ-PGA)	----	Films	Layer by-layer technique	----	L929 fibroblast	[128]
HA	ε-polylysine	Enzymatic	Hydrogels	Schiff base reaction	*E.coli* & *S. aureus*	L929 cells	[129]
HA	Oxidized hydroxyethylcellulose (OHEC)	EDC/HOBt and NaIO_4_	hydrogels	Schiff base reaction	----	NIH-3T3 cells	[130]
HA	Collagen	Horseradish peroxidase (HRP).	Hydrogel	In-situ coupling of phenol	*E.coli* & *S. aureus*	human microvascular endothelial cells (HMEC) and fibroblasts (COS-7)	[131]
HA	Various pH solutions of HAS	----	Hydrogel	freezing-thawing	----	White blood cell (WBC)	[132]
HA	Cornstarch	----	Films	Solvent-casting	*S. aureus, & E. coli*	L929 fibroblast	[133]
HA	de-acetylated LMW-HA (DHA) and the re-acetylated DHA (AHA)	----	hydrogels	Endotoxin Assay kit Instructions	----	HUVEC (Human Umbilical Vein Endothelial Cells)	[134]
HA	azobenzene and β-cyclodextrin	----	hydrogels	host-guest interactions	----	L929 cells	[135]

## Data Availability

Not applicable.

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
