# Peer review of "Recent Progress on Hyaluronan-Based Products for Wound Healing Applications"

_pharmaceutics, 2022, doi:10.3390/pharmaceutics14102235_

Round 1

Reviewer 1 Report

This manuscript covers the use of hyaluronic acid and its combined strategies for wound healing applications. It not only covers the functions of hyaluronic acid in the wound healing processes but also various strategies using HA and nanoparticles, particularly metallic nanoparticles. Since the purpose of wound healing is acceleration of the entire process, the use of metallic nanoparticles as antimicrobials can be beneficial. The majorities of the contents broadly cover the antimicrobial effects. However, no specifics are found why it is important to apply antimicrobials to promote the wound healing process. Thus, it would be better to have a paragraph or a few sentences explaining why it is important to have antimicrobial agents in the beginning of the manuscript. 

Other than this major point and a few minor typographical errors below, this is a well-prepared manuscript.

The minor errors noted are the following.

  1. Line 37, “healing is prolonged and problematic”. The prolonged healing will become problematic thus, “healing is prolonged becoming problematic”?

  2. Line 43, “However, wounds become chronic”. Based on the sentence, it may or may not become chronic as there are many factors. Should it be “wounds may become chronic”?

  3. Line 108, “maintain a moist atmosphere” and line 310 “maintain a moist environment” Be consistent with wording. “maintain a moist environment” is preferred.

  4. Line 259, the brackets are not closed properly. (Q Tang reported (2020))?

  5. Line 314, “therapeutic efficacy” instead of therapeutic efficiency”?

  6. Line 338, Table 1 caption. Table 1 only covers AgNPs. Should it be HA-based Ag nanoparticles? Furthermore, Should it be Base polymer instead of Based polymer in the first column? This applies to other tables.

  7. Line 385, References seem to be missing.

  8. Line 443, “HA” needs to be included as the sentence only shows AgNPs and GO. 

  9. Line 458, the subheading needs further details. “Drug loaded HA-nanocomposites” instead of “HA-drugs”.

  10. Line 571, a bullet point used in front of hydroxyl or alcohol functionality needs to be explained

  11. Line 716, “There are” instead of “it is”?

  12. Line 749, GO is not metallic nor particles. 

  13. Line 756, “antibacterial clay nanoparticles”? Do they come from natural clay or prepared in lab environments? Should it be covered in the previous sections if this is discussed as an active material?

  14. Line 786, should it be just “bacteria” not “bacterial agents”

Reviewer 2 Report

The manuscript needs to be revised urgently before it can be published.

1.There are problems with the journal names of the references. All references should be provided based on standard format.

2.The clarity of Figures 4, 5 and 7 needs to be improved.

3.The table should include a summary of wound healing.

4.There are some formatting errors, spelling errors and some incomprehensible places in the manuscript, such as line 284-289、365. It is recommended that the whole manuscript be carefully examined.

5.In situ should be italic in the text. Also use fixed format throughout the manuscript (in situ or in-situ).

6. Please unify the symbols of --------, …….., ….., etc. in the table.

7.Suggest to list abbreviations.

8.The theme of this review is wound healing, but the key information such as how to promote wound healing and the situation of wound healing are not fully described in the listed examples.

Reviewer 3 Report

The manuscript reported by Han and coworkers describes the use of composites of hyaluronic acid and metal nanoparticles such silver nanoparticles, gold nanoparticles and zinc oxide nanoparticles and their use in wound healing applications. This paper also describes the use of hyaluronic acid in drug delivery systems such antibiotics and other drugs also for the same applications.

The paper is well-written and covers the proposed subject. However, some considerations must be taking into account.

 As it is stated in the “instructions for authors” of Pharmaceutics “Acronyms/Abbreviations/Initialisms should be defined the first time they appear in each of three sections: the abstract; the main text; the first figure or table. When defined for the first time, the acronym/abbreviation/initialism should be added in parentheses after the written-out form.” In this way, I kindly ask that authors write-out form the first time the Acronyms/Abbreviations/Initialisms appears in the three sections.

Also, since this manuscript has plenty of abbreviations I suggest that a list of abbreviations in the end of the paper should be added, in order to be more reader-friendly.

Some phrases must be revised, it seems that a verb is missing in some of them. For example:

Pag 5, line 155: “The need to 155 understand the antibacterial mechanism in the wound healing process”

 pag 17, line 532: “Furthermore, nafcillin-loaded hydrogel characteristics such as 532 rheological, swelling studies, in vitro degradation, drug release kinetics, and in vitro 533 cytotoxicity”.

In 4.2 – Gold nanoparticles the references cited in are of 2012 and 2013. But in pag. 20 line 600 a more recent reference (2017) was added and discuss. In this way, it is strongly advised that this discussion should be added to the subtitle 4.2 Also please revise the bibliography search including more updated references for this subject.  For example: ACS Biomater. Sci. Eng. 2020, 6, 5132–5144; Applied Microbiology and Biotechnology 2018, 102, 4305–4318.

The description of future perspectives and challenges should be revised, it is not clear the content of this sub-title and its purpose.

Regarding the figures, the chemical structures must be increased and the same format should be added to all of those structures. Also, please increase the font size of the words that make up the figures.

Other revisions:

- please write microorganism instead of micro-organims

- please write in vivo, in vitro and et al.

- pag8 line 259 Q Tang reported (2020) seems “lost”

- pag 8 line 274, please do not add spaces in between-  5,10,15,20-tetrakis(4-methoxycarbonylphenyl)porphyrin

- please check reference n 41

Round 2

Reviewer 2 Report

1.Table 1 needs to be changed to three-line table.

2.The clarity of all the figures in the manuscript needs to be improved urgently.

3.There are some problems in the format of some references, such as 24,46, etc.

4.Italic expressions of in vivo and in vitro are required.
